# Potential Modulatory Effects of β-Hydroxy-β-Methylbutyrate on Type I Collagen Fibrillogenesis: Preliminary Study

**DOI:** 10.3390/ijms26199621

**Published:** 2025-10-02

**Authors:** Izabela Świetlicka, Eliza Janek, Krzysztof Gołacki, Dominika Krakowiak, Michał Świetlicki, Marta Arczewska

**Affiliations:** 1Department of Biophysics, Faculty of Environmental Biology, University of Life Sciences in Lublin, 20-950 Lublin, Poland; izabela.swietlicka@up.lublin.pl (I.Ś.); eliza.czarnecka295@interia.pl (E.J.); dominika.krakowiak@up.lublin.pl (D.K.); 2Department of Mechanical Engineering and Automation, Faculty of Production Engineering, University of Life Sciences in Lublin, 20-612 Lublin, Poland; krzysztof.golacki@up.lublin.pl; 3Department of Applied Physics, Faculty of Mechanical Engineering, Lublin University of Technology, 20-618 Lublin, Poland; m.swietlicki@pollub.pl

**Keywords:** collagen, β-hydroxy-β-methylbutyrate, atomic force microscopy, infrared spectroscopy, D-banding, fibrillogenesis

## Abstract

β-Hydroxy-β-methylbutyrate (HMB), a natural metabolite derived from the essential amino acid leucine, is primarily recognised for its anabolic and anti-catabolic effects on skeletal muscle tissue. Recent studies indicate that HMB may also play a role in influencing the structural organisation of extracellular matrix (ECM) components, particularly collagen, which is crucial for maintaining the mechanical integrity of connective tissues. In this investigation, bovine type I collagen was polymerised in the presence of two concentrations of HMB (0.025 M and 0.25 M) to explore its potential function as a molecular modulator of fibrillogenesis. The morphology of the resulting collagen fibres and their molecular architecture were examined using atomic force microscopy (AFM) and Fourier-transform infrared (FTIR) spectroscopy. The findings demonstrated that lower levels of HMB facilitated the formation of more regular and well-organised fibrillar structures, exhibiting increased D-band periodicity and enhanced stabilisation of the native collagen triple helix, as indicated by Amide I and III band profiles. Conversely, higher concentrations of HMB led to significant disruption of fibril morphology and alterations in secondary structure, suggesting that HMB interferes with the self-assembly of collagen monomers. These structural changes are consistent with a non-covalent influence on interchain interactions and fibril organisation, to which hydrogen bonding and short-range electrostatics may contribute. Collectively, the results highlight the potential of HMB as a small-molecule regulator for soft-tissue matrix engineering, extending its consideration beyond metabolic supplementation towards controllable, materials-oriented modulation of ECM structure.

## 1. Introduction

Skeletal muscle tissue, in addition to containing contractile muscle fibres, includes a substantial extracellular matrix (ECM) that provides essential structural and supportive functions [1]. ECM is primarily composed of collagen fibres, predominantly types I and III [2,3], which account for approximately 30% of the total protein mass in the human body and serve as a principal structural biomolecule in connective tissues such as skin, bone, tendon, cartilage, and muscle [4,5]. Its mechanical resilience derives from a unique supramolecular architecture: a right-handed triple helix composed of three left-handed polypeptide chains, rich in proline and hydroxyproline [6,7]. This helical structure spontaneously self-assembles into fibrils exhibiting a characteristic periodic banding pattern (D-periodicity) approximately 67 nm in length [8,9]. The folding and spontaneous aggregation of collagen monomers into fibrils are profoundly influenced by environmental factors and driven primarily by hydrophobic interactions, hydrogen bonding, and the establishment of salt bridges among collagen molecules [10,11]. Any modulation of these molecular interactions can critically impact collagen folding, aggregation dynamics, and, consequently, the mechanical and structural properties of the resulting fibrils.

Technological advances in areas such as the design of morphogenetic matrices and the development of collagen-based scaffolds for tissue engineering and regenerative therapies fundamentally rely on a detailed understanding of the relationship between the molecular structure of collagen and the properties of its supramolecular assemblies [12,13,14,15,16,17]. In recent years, increasing attention has been directed toward identifying naturally derived modulators capable of influencing collagen conformation, fibrillogenesis, and network organisation within biomaterials [18,19,20]. A detailed insight into the modification of collagen self-assembly is crucial for the rational design of next-generation collagen-based biomaterials with tailored structural and functional properties.

Small molecular compounds have been shown to modulate collagen biosynthesis at various levels. Some molecules, such as inhibitors of heat shock protein 47 (HSP47) or prolyl 4-hydroxylase, act by interfering with chaperoning or hydroxylation, thereby reducing collagen production and secretion, as observed in models of fibrotic disease [21,22]. Conversely, recent high-throughput screens have identified small molecules that can enhance collagen secretion by modulating ER () stress responses or vesicular trafficking [23]. Beyond intracellular regulators of biosynthesis, several small organic acids have been used to modulate collagen assembly and stability through predominantly non-covalent interactions. Prior studies reported that phytic acid alters collagen fibrillogenesis and network density in vitro [19]. More recently, citric acid has been used to crosslink collagen and improve the physicochemical properties of collagen sheets [20]. These findings highlight the feasibility of using small molecules as tools to either promote or attenuate collagen fibrillogenesis in a context-dependent manner.

Among bioactive molecules of natural origin, β-hydroxy-β-methylbutyric acid (HMB), a metabolite derived from the amino acid leucine, has gathered attention for its anabolic, anti-catabolic, and tissue-regenerative properties [24], as shown in Figure 1. Extensive studies over the past three decades, primarily conducted in athletic populations, have demonstrated the potential of HMB supplementation to increase muscle mass, strength, aerobic capacity, and resistance to fatigue by modulating anabolic and catabolic pathways [25]. As a short-chain carboxylic acid with a hydroxyl and methyl group at the β-position, HMB is capable of modulating cellular processes through hydrogen bonding, alterations in local dielectric properties, and steric effects on macromolecular structures. Specifically, HMB has been shown to enhance muscle protein synthesis via mammalian target of rapamycin (mTOR) activation [25], to suppress proteolytic pathways and concurrently attenuate protein degradation by inhibiting the ubiquitin-proteasome system [26] and apoptotic cascades [27]. Beyond its well-established anabolic properties, accumulating evidence suggests that HMB also exerts a significant influence on the metabolism and structural integrity of connective tissues. Studies in ageing and wound healing models have shown that HMB supplementation can accelerate collagen deposition and improve matrix remodelling [28]. Further, animal studies reported by Tomaszewska et al. demonstrated that maternal HMB administration modifies collagen architecture in developing cartilage and bone, with changes in fibre thickness and crosslinking patterns [29]. In our previous work, prenatal HMB supplementation was revealed to lead to significant changes in the distribution and structure of collagen and proteoglycans in articular cartilage [30]. Recent investigations have begun exploring innovative biomedical applications of HMB-loaded biomaterials, such as collagen I/ZnAl layered double hydroxide (LDH) composites, which promote sustained HMB release and offer enhanced potential for regenerating collagen-rich bone tissue [31].

Despite these observations, the precise molecular mechanisms underlying HMB’s modulatory role in the self-assembly of collagen on the molecular level have not been evaluated yet. To address this gap, an in vitro model of fibrillogenesis using bovine collagen type I is proposed to investigate the effect of HMB on the structure of collagen fibrils at two concentrations (0.025 M and 0.25 M). To comprehensively assess these potential effects, complementary structural and molecular analyses should be employed, combining nanoscale imaging with spectroscopic profiling to capture both morphological and conformational changes induced by HMB. Atomic force microscopy (AFM) enables high-resolution visualisation of individual type I collagen fibrils, providing quantitative data on fibril height, width, and the periodicity of D-bands, which is a hallmark of correctly assembled collagen. These parameters serve as critical indicators of the efficiency and fidelity of monomer aggregation into higher-order fibrillar structures. In parallel, Fourier-transform infrared (FTIR) spectroscopy aids in investigating molecular-level changes in the collagen secondary structure, with particular attention paid to the Amide I and III bands, which are sensitive to conformational changes within protein backbones. Second-derivative analysis of the Amide I region enables detailed sub-band deconvolution, providing insight into the stability of the triple helix and the overall integrity of the collagen matrix. These findings could address a critical gap in the current understanding of HMB’s extracellular roles, suggesting that it may act not only as a metabolic agent but also as a bioactive structural modulator of the extracellular matrix.

The objective of this study is to determine whether HMB directly interferes with collagen fibril and to evaluate the modulating effect on the self-assembly of type I collagen, using a simplified molecular model. We aim to mimic the extracellular matrix environment and assess whether HMB affects the stability and organisation of the triple helix. Our results will offer new insights into the rational design of regenerative therapies targeting collagen-based scaffolds.

## 2. Results and Discussion

Atomic force microscopy analysis facilitated the assessment of the physical structure of the studied collagen and the alterations induced by HMB throughout the fibrillogenesis process. Additionally, Fourier-transform infrared (FTIR) spectroscopy allowed for an examination of the structural organisation of the collagen and HMB molecules, as well as the molecular-level changes that occur.

### 2.1. Morphology

Collagen nanofibers require a specific environment to ensure the proper course of polymerisation, including, among others, an adequate concentration of potassium ions, a pH around 7.0, the presence of glycine, and an appropriate temperature. According to the literature [32], collagen fibres formed under such conditions should exhibit a width ranging from 50 to even 1000 nm and D-bands repeating approximately every 67 nm.

The collagen fibres from the control group were fibrils with an average width of approximately 358 nm, an average height of around 30 nm, and D-bands repeating roughly every 63 nm (Figure 1A, Table 1). The presence of a low concentration of HMB (0.025 M) during the polymerisation process did not cause any disruption to the collagen structure. As shown in Figure 1B and Table 1, the collagen fibres in this group had an average width of approximately 365 nm, an average height of 26 nm, and D-bands spaced at around 64 nm. It can be observed that compared to the control group, the spacing of the D-bands became more consistent, although the mean value did not differ significantly. In contrast, the average height of the fibres noticeably decreased, while their width remained at a similar level. Collagen in the form of fibres that polymerised in the presence of a higher HMB concentration (0.25 M) was characterised by a significantly reduced width and a markedly wider D-band compared to the control and HMB 0.025 groups (Figure 1C, Table 1).

D-bands, or transverse D-striations, form after the assembly of the quaternary structure through the aggregation and packing of tropocollagen molecules. Disruptions in the formation of D-bands, and thus in the quaternary structure or the breakdown of this structure, may result from environmental changes surrounding the collagen. These include variations in pH, temperature, and the degree of hydration. Additionally, exposure to various denaturing agents (both physical and chemical) can lead to the disintegration of collagen’s quaternary structure or prevent it from forming altogether [9].

In the present study, collagen incubation in the presence of HMB at a concentration of 0.025 M most likely disrupted the intermolecular hydrogen bonds involved in the formation of the triple-helix conformation, thereby inhibiting the formation of collagen fibres [33]. The reduction in fibril diameter suggests that HMB modulates the lateral growth of collagen fibrils through specific molecular interactions. HMB is a β-hydroxycarboxylic acid, possessing both a carboxylate group and a β-hydroxyl group in its structure. This dual functionality enables HMB to engage in electrostatic and hydrogen bonding interactions with collagen—something a simple carboxylic acid (lacking the β-hydroxyl group) cannot do as readily. It is proposed that HMB interacts with collagen monomers during fibrillogenesis via both electrostatic and hydrogen bonding mechanisms [34,35,36]. The deprotonated carboxylate of HMB can interact with positively charged amino acid residues on collagen, while the β-hydroxyl group can form hydrogen bonds with the polypeptide backbone or side chains of collagen. Such interactions may transiently “coat” the collagen monomers or form fibrils with HMB, influencing how collagen molecules pack together. If HMB occupies binding sites on the collagen surface or bridges between collagen molecules, it could hinder additional collagen from joining laterally, thus limiting the lateral assembly and producing thinner fibrils. In essence, the specific β-hydroxycarboxylate structure of HMB allows it to perturb collagen-collagen interactions in a targeted way. This behaviour would differ from that of structurally simpler acids, underscoring that the β-hydroxyl and the branched methyl of HMB confer a unique ability to modulate fibril formation.

A detailed understanding of the modulation of collagen self-assembly is crucial for the rational design of collagen-based biomaterials. Accordingly, secondary components, including glutaraldehyde, ionic liquids, and phytic acid (PA), have been employed to crosslink and stabilise collagen matrices [19,37,38]. PA likewise modulates collagen self-assembly, yielding markedly thinner fibrils while preserving the native D-periodicity (69–70 nm). At 1:1 (*w*/*w*) PA:collagen, nucleation accelerates and the network densifies, increasing gel stiffness; in excess, PA saturates hydrogen-bonding sites and imposes strong electrostatic repulsion, producing irregular, aggregated fibrils [19]. Thus, small molecules can either enhance fibrillogenesis in a coverage-dependent manner. Against this backdrop, the HMB response, characterised by thinner fibrils accompanied by a measurable D-spacing increase at high doses, suggests a distinct interfacial mechanism relative to PA.

This concentration-dependent modulation suggests a threshold effect, where low levels of HMB may subtly stabilise fibrillogenesis, whereas higher concentrations interfere with the spatial arrangement of collagen molecules. Although the system was buffered to physiological pH, the presence of high local concentrations of HMB may transiently reduce the pH in the vicinity of assembling collagen molecules, disrupting charge–charge and hydrogen-bond interactions essential for fibrillogenesis [39]. Moreover, HMB may compete with water for hydrogen bonding, interfering with collagen’s hydration shell and preventing the precise molecular alignment necessary for fibre assembly [40]. Finally, the accumulation of polar, low-molecular-weight solutes may alter the local dielectric environment or induce steric hindrance, impairing triple helix propagation [41]. These effects may collectively explain the morphological changes observed in collagen incubated with higher HMB concentrations.

### 2.2. Molecular Structure

The FTIR spectrum of type I collagen presented in Figure 2 is dominated by Amide bands and vibrations originating from amino acid side-chain groups [15,42,43]. The major features include the Amide I band, which mainly arises from the stretching vibrations in the peptide C=O backbone, appearing in the range of 1600–1700 cm^−1^. The Amide II band, dominantly attributed to the N–H bending coupled to C–N stretching vibrations, is located at 1500–1580 cm^−1^. The Amide III band, which is centred at approximately 1230–1240 cm^−1^, is mainly assigned to a complex of the stretching vibrations of C–N, the bending of N–H and wagging vibrations of CH_2_ groups in the glycine backbone and proline side chains. The bands found in the region between 3100 cm^−1^ and 3400 cm^−1^ arise due to O–H and N–H stretching of Amide A. There are also weaker bands arising from side-chain group vibrations, such as those of carboxylate groups (–COO^−^) at ~1400–1450 cm^−1^ (symmetric C–O stretching) and CH_2_ bending vibrations (~1450 cm^−1^), which correspond to the stereochemistry of the pyrrolidine rings of proline and hydroxyproline [42].

Upon adding HMB to collagen during fibrillogenesis, notable spectral shifts and intensity changes are observed in the Amide regions, reflecting modifications to collagen’s secondary structure and hydrogen-bonding network. The Amide A maximum is centred at 3314 cm^−1^ in a pure collagen sample with a slight shift to a higher wavenumber, and its intensity increases systematically with HMB concentration (Figure 2). Because the band frequency is sensitive to the position of hydrogen bonds, while its intensity reflects the number of N–H groups involved [43]. The nearly unchanged wavenumber, accompanied by increased absorbance, suggests that HMB introduces additional hydrogen bond donors or acceptors in the molecular surroundings of the collagen triple helix without significantly altering the original hydrogen bonding geometry. Alternatively, this effect may result from the overlap between the Amide A band and the O–H stretching vibrations originating from HMB.

In the Amide I region, all collagen-containing samples retain a prominent Amide I band, confirming that the polypeptide backbone remains in a folded (largely helical) conformation. The collagen itself exhibits an Amide I maximum at 1633 cm^−1^, which is within the range reported for native fibrillar collagens [44,45]. Notably, the 0.25 M HMB spectrum exhibits a slight upshift in the Amide I band relative to the control, with essentially the same position for 0.025 M HMB. A subtle increase in the Amide I frequency suggests a slightly stiffer C=O environment, but no large-scale unfolding occurs. For quantitative assessment, the wavenumber difference (Δυ) between the Amide I and Amide II bands was determined, and the results are presented in Table 2.

It is evident that all values were below 100 cm^−1^, indicating the preservation of the triple helix structure [44]. On the other hand, the Amide II band exhibits a measurable shift to higher wavenumbers of 6 and 10 cm^−1^ with increasing HMB concentration, respectively. Reduced N–H hydrogen bonding to the peptide backbone elevates the Amide II frequency [45], and HMB may compete for those H-bonds. Spectral overlap with HMB’s asymmetric COO^−^ stretch (1582 and 1544 cm^−1^) can pull the composite absorptions towards higher wavenumber, an effect documented when carboxylate vibrations overlap protein amide bands [34].

The changes are also seen in the Amide III region, and in the presence of HMB, this band at 1237 cm^−1^ appears to shift slightly toward higher wavenumber (by 4 cm^−1^). A shift in the Amide III to a higher frequency often signifies a slight loosening or distortion of the triple helix, since denatured or disordered collagen typically shows Amide III at elevated frequencies compared to native collagen. In our case, the 0.025 M HMB sample’s Amide III/1450 ratio is slightly higher than in pure collagen (Table 2). This indicates that the triple helix conformation is not destroyed by HMB, even at the highest concentration. Literature reports that an Amide III/1454 ratio of ~1.0 is typical for well-preserved helical structure [46,47]. In both HMB-containing samples, the ratio exceeded unity (1.203 ± 0.056 at 0.025 M; 1.093 ± 0.009 at 0.25 M). A ratio > 1 has previously been attributed to a preferential orientation or closer packing of helices within fibrils, which increases transition-dipole coupling in the Amide III mode [15].

In the spectral region between 1500 and 900 cm^−1^, characteristic for collagen side-chain vibrations, carbohydrate-associated modes, and C–O/C–N stretching, distinct differences were observed between pure collagen fibrils and those polymerised in the presence of HMB, as well as in relation to the spectrum of pure HMB. The bands at 1449 cm^−1^ and 1339 cm^−1^ are typically attributed to CH_2_ scissoring and COO^−^ symmetric stretching, respectively, arising from amino acid side chains such as proline and hydroxyproline [48]. For collagen fibrils formed in the presence of 0.025 M HMB, these bands underwent subtle changes related to a minor shift toward higher wavenumber, suggesting a more ordered packing of side chains and potentially a stabilising effect on the supramolecular fibril architecture. Conversely, at 0.25 M HMB, the intensities of these bands increased noticeably, particularly for the 1404 cm^−1^ band, indicating an altered hydration or electrostatic environment of the side-chain groups, consistent with structural perturbations induced by the increased contribution from carboxylate groups [49]. In the range of 1339–1234 cm^−1^, which corresponds primarily to CH_2_ wagging, N–H bending, and C–N stretching of Amide III modes, collagen fibrils polymerised with HMB concentration showed a band pattern closely resembling that of the control sample. This suggests that the triple-helical conformation remained largely intact, in agreement with earlier reports that organic acid additives at low concentrations did not disrupt collagen’s native folding [19,20].

A prominent alteration was observed in the spectra of the 0.25 M HMB–collagen sample in the region between 1120 and 1020 cm^−1^. Vibrations in the region of 1020–1030 cm^−1^ are typically assigned to the C–O stretching of alcohol or hydroxy-acid groups engaged in strong hydrogen bonding [50]. While control collagen and the 0.025 M HMB sample exhibited only a modest band at 1078 cm^−1^ and 1031 cm^−1^, which arise from the stretching (C–O) and (C–O–C) absorptions of the carbohydrate moieties [42], a higher HMB condition produced two significant features: an intense, newly emergent band at 1114 cm^−1^ and a pronounced enhancement of the 1023 cm^−1^. The latter is shifted from a weak feature at 1017 cm^−1^ and aligns closely with the C–O stretching band observed in the pure HMB spectrum, suggesting direct incorporation of HMB-specific vibrational modes into the composite fibril spectrum. The 6 cm^−1^ shift towards higher wavenumber and marked intensity gain therefore point to incorporation of HMB’s C–OH / COO^−^ functionalities into the collagen network, probably via hydrogen bonding or ion-pairing with cationic sites on collagen, thereby modifying the vibrational landscape of the carbohydrate and backbone region [35]. Such interactions would enhance the overall hydrogen-bond density, consistent with the increased intensity of the Amide A band.

Overall, the FTIR analysis reveals that HMB interacts with collagen in a way that modulates hydrogen bonding and secondary structure without destroying the triple helix. We interpret these findings as suggesting that HMB becomes incorporated among the collagen fibrils during polymerisation, where it can form additional hydrogen bonds with collagen (especially at higher concentrations) and likely engages in electrostatic interactions. Indeed, prior studies have shown that introducing small polyhydroxy or polyacid molecules can influence collagen’s FTIR bands and stability, and covalent cross-linkers are known to shift amide peaks as new bonds form [51,52]. Similarly, Andonegi et al. reported on the interactions of collagen with citric acid at varying concentrations, demonstrating that collagen stabilisation was achieved through hydrogen bonding. Although no evidence of chemical crosslinking was observed, owing to the absence of significant alterations in the Amide region, physical crosslinking was clearly detected in FTIR spectra [20]. The concentration-dependent impact of organic acids on collagen assembly was also investigated by Tu et al., who demonstrated that high concentrations of phytic acid promote the formation of a denser collagen fibril network [19]. This effect was attributed to the occupation of most of the hydrogen-bonding sites on collagen by phytic acid molecules, leading to electrostatic repulsion between the bound acid residues. Such repulsion was proposed to hinder both intra- and inter-molecular interactions within the collagen matrix, thereby altering its structural organisation. Overall, the FTIR analysis reveals that HMB interacts with collagen in a way that modulates hydrogen bonding and secondary structure without destroying the triple helix. These findings rationalise the morphologies observed by AFM, which show more orderly fibrils at low HMB and perturbed packing at high HMB, and position HMB as a bio-compatible additive for tailoring extracellular matrix architecture. This is consistent with HMB’s known biological role in improving connective tissue integrity and healing, where it might promote a more cross-linked or finer ECM network.

To interpret the effect of HMB as a potential modulator of collagen structure, particularly through the analysis of the Amide I band, attention should be focused on the changes in the relative contributions of secondary structure components. The curve-fitting procedure applied to the 1700–1600 cm^−1^ spectral region (Figure 3) enabled the determination of the secondary structure components of collagen, as previously described [9,53,54,55].

Analysis of the Amide I band shape, using second derivative minima to determine the position of the components, reveals a concentration-dependent change in the equilibrium of secondary structure populations in collagen (Figure 3A) formed from HMB (Figure 3B,C). The control spectrum (Figure 3A) shows the known sub-band distribution characteristic of type I fibrous collagen that includes β-sheets (1691 cm^−1^, 1679 cm^−1^, 1626 cm^−1^), β-turns (1668 cm^−1^, 1608 cm^−1^), α-helices (1658 cm^−1^), triple helices (1637 cm^−1^), and random coil structures (1647 cm^−1^). As summarised in Table 3, incorporation of HMB into the system was accompanied by an increase in α-helix content. This rise suggests that HMB promotes the stabilisation of α-helical domains, consistent with conformational adjustments arising from HMB–collagen non-covalent interactions, in agreement with the subtle modifications previously detected in the overall Amide region.

More specifically, incorporation of 0.025 M HMB causes the intensity to be redistributed away from the random-coil/β-turn region toward components assigned to triple-helix and α-helical order, with a modest reinforcement of β-sheet-like packing. This pattern, together with the preserved Amide III/1450 cm^−1^ integrity index reported for the same samples, may be interpreted as stabilisation of the native triple helix and tightening of interchain hydrogen bonding at low HMB content. One possible interpretation is that HMB alters local hydrogen-bonding networks and hydration around peptide groups during assembly. Electrostatic contributions cannot be excluded, but with the present methods, we cannot distinguish them from alternative non-covalent scenarios. Such interactions would lower local conformational entropy and promote helix maintenance during assembly, consistent with the more regular fibrillar morphologies observed by AFM at low dose (Figure 1). On the other hand, at a concentration of 0.25 M HMB, the proportion of α-helices and β-sheets is further enhanced, the population of β-turns is reduced, and the triple helix component relaxes to levels similar to those in the control, despite the overall predominance of helices. HMB promotes a redistribution from turns to α-helical and β-sheet-like arrangements, as well as enhanced interchain packing. This interpretation aligns with the sensitivity of Amide I sub-bands to hydrogen-bond topology and interchain organisation in fibrous proteins, while acknowledging the intrinsic non-uniqueness of band-shape analyses in collagen [56]. Notably, the triple-helix fraction remained relatively stable across all conditions, suggesting that HMB did not disrupt the native collagen backbone but rather modulated its supramolecular organisation. The observed reduction in disordered motifs (random coils, β-turns) in favour of more ordered elements (α-helices, β-sheets) implies that HMB promotes intramolecular hydrogen bonding and enhanced chain alignment, potentially leading to increased mechanical stability and resistance to denaturation. These findings align with previous reports that some natural modulators can influence collagen self-assembly by stabilising ordered structures without compromising triple-helical integrity [20,46,57].

## 3. Materials and Methods

### 3.1. Collagen and β-Hydroxy-β-Methylbutyric Acid (HMB)

The research material consisted of a Type I collagen solution in monomeric form (PureCol^®^-S, Advanced BioMatrix, Carlsbad, CA, USA) at a concentration of 3 mg/mL, maintained at a pH range of 1.9–2.3, and stored at 4 °C. HMB in powder form was obtained from BULK POWDERS™ (Bulk, Colchester, Essex, UK). The manufacturer guarantees that it is 100% β-hydroxy-β-methylbutyric acid. The HMB powder was accurately weighed and dissolved in distilled water to prepare solutions at two different concentrations: 0.025 M (HMB 0.025) and 0.25 M (HMB 0.25). The upper concentration was chosen to remain below the solubility limit of HMB in water (~0.32 M) [58] and to test a disruptive regime of collagen self-assembly. The lower concentration was chosen based on the research regarding molecular systems, where the influence of HMB on collagen assembly was studied [20,31]. The prepared solutions were stored in a refrigerator for a maximum of 2–3 weeks. Before starting the experiment, both substances were analysed using an FTIR spectrometer to verify the compliance of the obtained spectra with those presented in the literature.

### 3.2. Buffer Solution

Collagen synthesis is sensitive to various factors, including the presence of specific ions, pH level, and temperature. Therefore, a buffer solution was prepared to ensure suitable conditions for polymerising the applied collagen monomer. The buffer solution was composed of:Phosphate-buffered saline (PBS, Sigma-Aldrich, Saint Louis, MO, USA) in tablet form, dissolved in 200 mL of distilled water (Millipore, Bedford, MA, USA). Dissolving one PBS tablet in 200 mL of deionised water resulted in a 0.01 M phosphate buffer, 0.0027 M potassium chloride, and 0.137 M sodium chloride at a pH of 7.2–7.6 at 25 °C.Potassium chloride (KCl, Sigma-Aldrich, Saint Louis, MO, USA) at a concentration of 200 mM.Glycine (Sigma-Aldrich, Saint Louis, MO, USA) at a concentration of 50 mM.

Mentioned components were mixed in a 2:1:1 ratio, respectively. At each stage of buffer preparation, the pH level was monitored and adjusted by adding small amounts of either sodium hydroxide (NaOH, Sigma-Aldrich, Saint Louis, MO, USA) or 0.1 M hydrochloric acid (HCl, Sigma-Aldrich, Saint Louis, MO, USA) as needed to maintain a physiological pH range of 7.0–7.3.

### 3.3. Sample Preparation and Atomic Force Microscopy (AFM) Analysis

Samples were prepared following a modified protocol from [59]. The buffer solution and collagen were removed from the refrigerator 30 min before the preparation. After the room temperature was reached, 36 µL of buffer was added to 1.5 mL Eppendorf tubes, followed by 4 µL of collagen monomer. Depending on the sample type, 10 µL of distilled water (control) or an HMB solution at one of two concentrations (0.025 M and 0.25 M) was added. The samples were vortexed (Vortex mixer, Labnet, Carry, NC, USA) until a clear and colourless solution was obtained. The prepared samples were incubated at 37 °C for 3.5 h (AccuTherm Thermostat, Labnet, Cary, NC, USA). After incubation, 20 µL of the collagen solution was deposited onto the surface of a cleaned mica sheet (Agar Scientific, Rotherham, UK). After 5 min, the mica was rinsed with distilled water (Millipore, Bedford, MA, USA) to remove excess fibres. The mica surface was then dried with nitrogen. The collagen samples were evaluated using an optical microscope (DM 2500, Leica Microsystems, Wetzlar, Germany) and examined with an atomic force microscope (INTEGRA Prima, NT-MDT, Moscow, Russia), following the method described in [59].

The AFM was equipped with RTSPA-300 probes (Bruker, Berlin, Germany), featuring a length of 125 μm, a thickness of 4.0 μm, a tip radius of 10 nm (Figure 1A), and an average resonance frequency of 310 kHz (Figure 1B). The device operated in non-contact mode. All images were captured at a low scanning speed (1 Hz) and a resolution of 512 × 512 pixels per scan. For each group, three independent samples were prepared and scanned. At least 30 fibres per group were analysed to ensure a representative morphological assessment. Only well-separated, unbundled fibrils were included in the measurements. Fibril-level measurements were averaged within each biological replicate, and only these replicate means were used in the statistical analyses. The physical parameters of the collagen fibres, including width, height, and periodicity of D-bands, were determined based on the obtained images, while care was taken to ensure that the tested collagen fibres were similarly oriented [60]. Image analysis was conducted using Gwyddion 2.51 (open-source software under the GNU General Public licence).

### 3.4. Fourier Transform Infrared (FTIR) Analysis

Samples for Fourier-transform infrared (FTIR) spectroscopy were prepared following the same protocol as for AFM analysis. After incubation, the suspensions were centrifuged to sediment the newly formed collagen fibrils, and the supernatant was carefully removed. The pelleted material was gently dried under a stream of nitrogen gas and subsequently subjected to spectroscopic analysis using an IRSpirit FTIR spectrometer (Shimadzu, Kyoto, Japan) equipped with a QATR-S single-reflection accessory. Spectra were acquired over the range of 4000–500 cm^−1^ at a spectral resolution of 4 cm^−1^, with each spectrum representing the average of 40 scans. For each experimental condition, three independently prepared samples were analysed, and each sample was measured in triplicate, yielding a total of nine spectra per group. Background spectra, recorded from a clean ATR crystal under identical acquisition settings, were automatically subtracted from the sample spectra to eliminate contributions from atmospheric water vapour and CO_2_. Spectra were averaged within each biological replicate, and these biological means (n = 3) were used for statistical analyses. Spectral preprocessing included polynomial baseline correction, “offset correction” transformation, smoothing using a 9-point Savitzky–Golay algorithm, and area normalisation within the 1750–1475 cm^−1^ region. The integrity of the collagen triple helix was assessed by calculating the absorbance ratio between the Amide III band at 1235 cm^−1^ and the band at 1450 cm^−1^, the latter associated with the stereochemistry of pyrrolidine rings in proline and hydroxyproline residues—an established metric for monitoring triple helix preservation [61]. Second-derivative spectra of the Amide I region were employed as positional constraints for curve-fitting analysis to quantify changes in collagen secondary structure. All spectral processing and quantitative analyses were performed using GRAMS/AI 8.0 (Thermo Galactic, Waltham, MA, USA), Matlab 2025a (MathWorks Inc., Natic, MA, USA) and OriginPro 2022 (OriginLab Corporation, Northampton, MA, USA).

### 3.5. Study Limitations

While the results of this study clearly demonstrate the structural effects of HMB on collagen morphology and secondary structure, certain limitations should be acknowledged. FTIR spectroscopy is inherently sensitive to residual moisture content, local film thickness, and minor inconsistencies in sample drying. Although measures were taken to minimise these factors, minor baseline variations may still influence peak area ratios, particularly for Amide bands. However, the consistency of spectral shifts across biological replicates and their alignment with AFM-derived morphological changes support the conclusion that these observations reflect true structural transformations rather than artefacts. To cross-validate these findings with methods less affected by hydration, future work will include temperature-controlled circular dichroism, AFM-Raman mapping, and solid-state NMR.

Only two concentrations above physiological levels (0.025 M and 0.25 M) were examined rather than a full dose–response series. To define the dose–response shape, we will implement a densified, constant-ionic-strength series, extending from the micromolar and sub-millimolar range up to 0.25 M, together with time-course assays (e.g., turbidity kinetics) to distinguish effects of HMB on nucleation, fibril growth, and late-stage stabilisation. The present study did not measure ζ-potential or local pH at the fibril interface. To directly probe our non-covalent interaction hypothesis, follow-up experiments will include ζ-potential titrations of fibril suspensions at pH 7.4.

### 3.6. Statistical Analysis

The collected data were analysed in terms of their nature. Data points identified as potential outliers were verified for biological plausibility and retained in the dataset unless clear evidence of measurement error was found. The assumptions of normal distribution and variance homogeneity were confirmed by the Shapiro–Wilk test and Levene’s test, respectively. A one-way analysis of variance (ANOVA) was used to evaluate changes in both collagen fibre morphology (width, height, D-banding) and FTIR-derived spectral characteristics (Δυ, AmideIII/1450 cm^−1^ ratio and Amide I secondary structure). Post hoc Tukey’s tests were conducted to determine the nature of differences between groups. Model residuals were tested to validate the assumption of normality and homoscedasticity using QQ plots, the Shapiro–Wilk test, residual–fitted values plots and Bartlett’s test. When assumptions were not met, nonparametric analysis was conducted. For all analyses, *p* < 0.05 was considered statistically significant. Data were presented as least squares (LS) means with standard deviations (SD) or medians with interquartile ranges (IQR). For all inferential analyses, the experimental unit was the biological replicate (n = 3 independent sample preparations per group). Technical replicates (≥30 fibrils or 3 spectra per group) were averaged within each biological replicate to avoid pseudoreplication. Only these biological means were entered into the ANOVA/Tukey tests.

The RStudio (2023.09.1, build 494, Posit Software, PBC, WA, USA), Statistica 13 (TIBCO Software Inc., Palo Alto, CA, USA), and OriginLab 2022 (OriginLab, MA, USA) software were used for the statistical analyses.

## 4. Conclusions

The presented work demonstrates that β-hydroxy-β-methylbutyrate directly modulates type I collagen self-assembly in a concentration-dependent manner. Polymerisation in the presence of low HMB (0.025 M) produced fibrils with a more regular ultrastructure and spectroscopic evidence of reinforced native order consistent with an increased contribution of ordered helical motifs. By contrast, high HMB (0.25 M) produced pronounced morphological perturbations (reduced fibril dimensions and enlarged D-periodicity) accompanied by a redistribution of Amide-I sub-bands toward α-helical/β-sheet-like contributions, a spectral shift in Amide II towards higher wavenumber, growth of carboxylate band, and the emergence of a distinct region indicative of interfacially bound β-hydroxycarboxylate. Based on the observed amide-band redistribution and features in the region reflecting hydroxylated side chains in collagen and β-hydroxycarboxylate contributions from HMB, we suggest that short-range electrostatic interactions (ion pairing) and hydrogen-bond mediation may contribute to the effect. At low HMB concentration, the changes are subtle and compatible with a preserved triple helix and slightly tighter interchain contacts. At high HMB concentrations, fibrils are thinner yet exhibit greater self-assembly, consistent with the formation of a denser fibrillar network.

These results suggest that HMB is a non-covalent modulator of collagen architecture, enabling controllable fibril organisation in collagen-based biomaterials and formulation development.

## Data Availability

Data reported in this manuscript will be available upon request.

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
