# Peer review of "Potential Modulatory Effects of β-Hydroxy-β-Methylbutyrate on Type I Collagen Fibrillogenesis: Preliminary Study"

_ijms, 2025, doi:10.3390/ijms26199621_

Round 1

Reviewer 1 Report

Comments and Suggestions for Authors

How do you ensure that the experimental unit in the inferential analyses is the biological sample (n = 3) rather than the individual fibrils (≥ 30 technical replicates)? Could you make this explicit in the Methods/Statistics section?

You propose a surface-binding mechanism (ion pairing and H-bonding). Do you have ζ-potential data, ionic strength/local pH measurements, or controls that support this hypothesis?

Were the statistical tests described (ANOVA/Tukey, etc.) also applied to the FTIR variables (Δν, Amide III/1450 ratio, Amide I fractions)?

In Table 2, could you add 95% CIs (or SD) in addition to SE to assess the absolute variability of the spectral measurements?

      Comments on the Quality of English Language

How do the authors ensure that the experimental unit in the inferential analyses is the biological sample (n = 3) rather than the individual fibrils (≥ 30 technical replicates)? Could you make this explicit in the Methods/Statistics section?

the authors propose a surface-binding mechanism (ion pairing and H-bonding). Do you have ζ-potential data, ionic strength/local pH measurements, or controls that support this hypothesis?

Were the statistical tests described (ANOVA/Tukey, etc.) also applied to the FTIR variables (Δν, Amide III/1450 ratio, Amide I fractions)?

In Table 2, could the authors add 95% CIs (or SD) in addition to SE to assess the absolute variability of the spectral measurements?

Author Response

We sincerely thank Reviewer#1 for the careful evaluation of our manuscript and for the insightful comments that helped us improve its clarity and methodological transparency. Please find our detailed responses to each point below. For clarity, line numbers refer to the revised version (no tracked changes).

 1. How do you ensure that the experimental unit in the inferential analyses is the biological sample (n = 3) rather than the individual fibrils (≥ 30 technical replicates)? Could you make this explicit in the Methods/Statistics section?

We thank the Reviewer for this important remark. We clarified in the Materials and Methods section that the experimental unit in all inferential analyses was the biological replicate (n = 3 independent preparations per group). Apart from that, each table caption was enriched with a sentence explaining the presented values (biological replicates and the number of measurements taken within it).

  • Sample Preparation and Atomic Force Microscopy (AFM) Analysis (Lines 455-456): “Fibril-level measurements were averaged within each biological replicate, and only these replicate means were used in the statistical analyses”
  • Fourier Transform Infrared (FTIR) Analysis (Lines 473-475): “Spectra were averaged within each biological replicate, and these biological means (n = 3) were used for statistical analyses.”
  • Statistical analysis (Lines 499-503): “For all inferential analyses, the experimental unit was the biological replicate (n = 3 independent sample preparations per group). Technical replicates (≥30 fibrils or 3 spectra per group) were averaged within each biological replicate to avoid pseudoreplication. Only these biological means were entered into the ANOVA/Tukey tests.”
  1. You propose a surface-binding mechanism (ion pairing and H-bonding). Do you have ζ-potential data, ionic strength/local pH measurements, or controls that support this hypothesis?

Thank you for raising this important point. We did not obtain ζ-potential data in the present study. We now explicitly acknowledge this in the Study Limitations and, in future work, will include ζ-potential titrations of fibril suspensions at pH 7.3 (with constant-ionic-strength controls and local-pH data) to directly probe surface-charge effects. To minimise confounds, our experiments were designed to keep bulk pH and ionic strength constant across all conditions and to read out structural/morphological consequences after fibrillogenesis. A solution was prepared with 10 mM PBS buffer/200 mM KCl/ 50 mM glycine and then titrated to a physiological pH (7.3), which was described in detail in section 3.2. Given HMB’s pKₐ = 4.4,  HMB is almost entirely in the anionic form at pH 7.3, so the observations cannot be attributed to acidification of the sample by HMB. In addition, the ATR-FTIR method was performed on pelleted fibrils after removing the supernatant, which reduces contributions from unbound solute. Under these controlled conditions, the results from the AFM and FTIR techniques reveal concentration-dependent changes that are consistent with non-covalent modulation (electrostatics and H-bonding), but are not diagnostic of a single mode of interaction.

We understand the concern that, in the absence of dedicated assays localising binding sites, a surface-binding mechanism should be used with caution. In fact, the mechanism we propose is a hypothesis based on observations of morphology (AFM) and structural changes (detailed analysis of FTIR spectra). However, it has been pointed out that the effect is concentration-dependent and is particularly evident in the observed amide-band redistribution and features in the region reflecting hydroxylated side chains in collagen, as well as hydroxycarboxylate contributions from HMB. We advanced a working hypothesis that short-range electrostatic interactions (ion pairing) and hydrogen-bond mediation may contribute to the effect.

Therefore, we have revised the manuscript in several places to address the Reviewer's comments. We hope the updates satisfactorily address the points raised.

Namely, the edited passage in the Abstract appears at lines 26-31

We revised the Introduction in line with the reviewer’s comment; see lines 121-126.

We softened interpretative language, removed some unconfirmed statements in the Results and Discussion (Lines  180-183; 379-382), and in the Conclusions (Lines  537-542).

  1. Were the statistical tests described (ANOVA/Tukey, etc.) also applied to the FTIR variables (Δν, Amide III/1450 ratio, Amide I fractions)?

Thank you for the comment. ANOVA followed by Tukey’s post hoc test was applied to both AFM-derived morphological parameters and FTIR-derived spectral characteristics (Δν, Amide III/1450 ratio, and Amide I secondary structure fractions). We have revised the Statistical Analysis subsection to make this explicit (Lines 491-493): “A one-way analysis of variance (ANOVA) was used to evaluate changes in both collagen fibre morphology (width, height, D-banding) and FTIR-derived spectral characteristics (Δυ, AmideIII/1450cm-1 ratio and Amide I secondary structure).”

  1. In Table 2, could you add 95% CIs (or SD) in addition to SE to assess the absolute variability of the spectral measurements?

We appreciate this suggestion. To provide a more transparent view of variability, we revised all tables to present data consistently as mean ± SD. This change applies to Tables 1–3. The corresponding captions and the Statistical Analysis section were updated accordingly.

Reviewer 2 Report

Comments and Suggestions for Authors

The author's research indicates that β-hydroxy-β-methylbutyrate (HMB) directly affects the self-assembly of type I collagen in a concentration-dependent manner. When polymerization occurs in the presence of low concentrations of HMB (0.025 M), the resulting fibrils exhibit a more regular ultrastructure. Spectroscopic analyses reveal a reinforced native order, characterized by a maintained Amide III/1450 cm−1 integrity index and Amide I band shapes that suggest an increased presence of ordered helical structures.

In contrast, high concentrations of HMB (0.25 M) lead to significant morphological changes, characterized by reduced fibril dimensions and enlarged D-periodicity. According to the authors, this is accompanied by a shift of Amide-I sub-bands toward α-helical and β-sheet-like contributions, an upward spectral shift of Amide II toward higher wavenumbers, an increase in the carboxylate band around 1400 cm−1, and the appearance of a distinct band at 1114 cm−1, which indicates the presence of interfacially bound β-hydroxycarboxylate.

Notably, the Δυ (Amide I–Amide II) value remained below 100 cm−1 across all conditions, suggesting that the triple-helical structure of collagen is maintained despite the supramolecular rearrangement. The authors' spectroscopic analyses and atomic force microscopy (AFM) data support a surface-binding mechanism in which HMB interacts with collagen through carboxylate ion pairing and β-hydroxyl hydrogen bonding.

At low interfacial coverage, these interactions strengthen interchain contacts and stabilize the natural fibrillogenesis process. However, at high coverage, they compete with peptide–peptide hydrogen bonds, altering local hydration and promoting lateral packing, which imparts a β-sheet-like character to the structure. At the same time, the overall helicity remains largely intact. Interestingly, at high HMB concentrations, the fibrils are thinner but demonstrate enhanced self-assembly, indicating the formation of a denser fibrillar network.

The strength of the manuscript: a high degree of novelty, namely the authors proved that in addition to HMB being a regulator of metabolic reactions, HMB directly affects the structure of biomolecules, i.e., to the collagen self-association process.

Weakness of the manuscript: The authors could have included a series of small carboxylic acids where they vary the length of the C chain, the number of OH groups, and the absence of a methyl group in the β-position.

Recommended minor corrections:

  1. The authors could give the structure of HMB in the introductory part.
  2. Schematically show the introduction from lines 73 to 93.
  3. Are there known small molecules with similar properties to HMB in terms of collagen association?
  4. Explain how the concentrations of 0.025 M and 0.25 M were chosen, whether they are correlated with physiological concentrations, why more concentrations between the two end values ​​were not taken, in order to more accurately determine the concentration-dependent action.
  5. Does the methyl group in the β-position of HMB have any significance in the described mechanism?

Author Response

We sincerely thank the Reviewer for the thorough and thoughtful evaluation of our work. We greatly appreciate the recognition of the novelty and relevance of our study, as well as the constructive comments and suggestions for additional explanations, schematic presentation, and further elaboration of our considerations. Please find our detailed responses to each point below. For clarity, line numbers refer to the revised version (no tracked changes).

  1. The authors could give the structure of HMB in the introductory part
  2. Schematically show the introduction from lines 73 to 93.

Thank you for the remark. The scheme presenting both the chemical structure and reported HMB mechanism of actions was added to the Introduction section as proposed (Scheme 1).

  1. Are there known small molecules with similar properties to HMB in terms of collagen association?

Thank you for pointing this out. Several low-molecular compounds are reported to modulate collagen assembly non-covalently, in ways that are conceptually similar to our interpretation for HMB. Our Introduction (Lines 66-71) and Results and Discussion (Lines 192-199, 323-336) already contextualise HMB alongside small, polyfunctional acids that non-covalently modulate collagen assembly. For example, citric acid stabilises collagen sheets via hydrogen bonding without forming covalent crosslinks, as shown by Andonegi et al. (2020. Likewise, phytic acid (a polyacid) modulates collagen self-assembly and can yield thinner fibrils and denser networks depending on dose (see Tu et al.,2008).

We have also added the following content to the Introduction because we reported it only in the Results and Discussion: “Beyond intracellular regulators of biosynthesis, several small organic acids have been used to modulate collagen assembly and stability through predominantly non-covalent interactions. Prior studies reported that phytic acid alters collagen fibrillogenesis and network density in vitro [19]. More recently, citric acid has been used to crosslink collagen and improve physicochemical properties of collagen sheets  [20]”. (Lines 66-71)

  1. Explain how the concentrations of 0.025 M and 0.25 M were chosen, whether they are correlated with physiological concentrations, why more concentrations between the two end values ​​were not taken, in order to more accurately determine the concentration-dependent action.

Thank you for pointing this out, and we agree that this required clearer justification. Our aim was to bracket two regimes in a controlled, buffered, simple molecular model: a low-concentration condition (0.025 M), where subtle effects might appear, and a high-concentration one (0.25 M), where interference with assembly becomes evident, as studied by the AFM and FTIR methods. The system was buffered at pH 7.3 with a defined ionic strength (PBS/KCl/glycine) to separate local assembly effects from overall pH/ionic drift. Accordingly, 0.025 M HMB was selected to be sub-dominant relative to the principal salts (comparable to minor buffer components), whereas 0.25 M HMB was chosen to be comparable to above the monovalent ionic background in the buffer. It should be emphasised that these concentrations are not meant to mimic plasma levels. These doses are intentionally above the physiological level and serve the study of interactions on the molecular level rather than an exposure mode. In humans, fasting plasma HMB is typically ~2–5 µM, and even with supplementation, peaks are generally sub-millimolar, orders of magnitude below our test window. Thus, our experiment was designed to follow common practice in collagen self-assembly studies, where above physiological levels are used to isolate the mechanism. Use of these HMB concentrations to probe non-covalent modulation of collagen is consistent with prior work on citric acid and phytic acid reported by Andonegi et al. 2020 and Tu et al. 2018, respectively, which alter collagen assembly without covalent crosslinking. Moreover, the upper HMB level was set below the aqueous solubility limit of the free-acid form because the reported aqueous solubility of the free-acid form of HMB is > 38 g/L (≈0.32 M).

Explanation supported by the new literature position [58] was added to the Materials and Methods section, Collagen and β-Hydroxy-β-Methylbutyric Acid (HMB) subsection (Lines 409-413): “The upper concentration was chosen to remain below the solubility limit of HMB in water (~0.32M) [58] and to test a disruptive regime of collagen self-assembly. The lower concentration was chosen based on the research regarding molecular systems, where the influence of HMB on collagen assembly was studied [20, 31]”.  Additionally, in the Study Limitations section, it was explained that further studies are planned with different HMB concentrations (Lines 519-526).

  1. Does the methyl group in the β-position of HMB have any significance in the described mechanism?

We appreciate this insightful question. We did not vary HMB’s substituents in this study, so we avoid a definitive claim. Probably, the β-methyl could influence steric fit and local hydrophobicity at collagen surfaces and slightly alter hydration around the β-hydroxycarboxylate, however, based on current data, we abstain from assigning it a specific role. On the other hand, FTIR spectra cannot reliably isolate the role of the β-methyl group. Signals specific to –CH₃ overlap with the intense collagen’s own -CH bands and are generally insensitive to weak, short-range interactions. In our case, we focused on the fingerprint region, which better reflects the hydration/hydrogen bonds of peptide groups and the contribution of β-hydroxycarboxylate, rather than the position of the methyl itself. Therefore, the FTIR method does not allow us to isolate the contribution of the β-methyl group here.

Reviewer 3 Report

Comments and Suggestions for Authors

The manuscript provides relevant data through an extensive approach. Please highlight better in the methodology how the two concentrations of HMB (0.025 M and 0.25 M) were determined or selected by the Authors and add references, if applicable.

Author Response

We thank the Reviewer for the positive assessment of our manuscript and for the constructive remarks aimed at improving its clarity and presentation. We have addressed the request to better justify the choice of HMB concentrations in the Materials and Methods, with added explanation and references, and we provide detailed responses to all other comments below. For clarity, line numbers refer to the revised version (no tracked changes).

  1. The manuscript provides relevant data through an extensive approach. Please highlight better in the methodology how the two concentrations of HMB (0.025 M and 0.25 M) were determined or selected by the Authors and add references, if applicable.

Thank you for pointing this out, and we agree that this required clearer justification. Our aim was to bracket two regimes in a controlled, buffered simple molecular model. The system was buffered at pH 7.3 with defined ionic strength (PBS/KCl/glycine) to separate local assembly effects from overall pH/ionic drift.

Accordingly, 0.025 M HMB was selected to be sub-dominant relative to the principal salts comparable to minor buffer components, namely 10 mM PBS buffer, 200 mM KCl and 50 mM glycine). Whereas HMB concentration of 0.25 M was chosen to be comparable to or above the monovalent ionic background in the buffer. It should be emphasised that these concentrations are not meant to mimic plasma levels. These concentrations are intentionally above the physiological level and serve the study of interactions on the molecular level rather than an exposure mode. In humans, fasting plasma HMB is typically ~2–5 µM, and even with supplementation, peaks are generally sub-millimolar, orders of magnitude below our test window. Thus, our experiment was designed to follow common practice in collagen self-assembly studies, where above physiological levels are used to isolate the mechanism. Use of these HMB concentrations to probe non-covalent modulation of collagen is consistent with prior work on citric acid and phytic acid, which alter collagen assembly without covalent crosslinking. Moreover, the upper HMB level was set below the aqueous solubility limit of the free-acid form because the reported aqueous solubility of the free-acid form of HMB is > 38 g/L (≈0.32 M).

Explanation supported by the new literature position [58] was added to the Materials and Methods section, Collagen and β-Hydroxy-β-Methylbutyric Acid (HMB) subsection (Lines 409-413): “The upper concentration was chosen to remain below the solubility limit of HMB in water (~0.32M) [58] and to test a disruptive regime of collagen self-assembly. The lower concentration was chosen based on the research regarding molecular systems, where the influence of HMB on collagen assembly was studied [20, 31]”.  Additionally, in the Study Limitations section, it was explained that further studies are planned with different HMB concentrations (Lines 519-526).

Reviewer 4 Report

Comments and Suggestions for Authors The authors present a purely in vitro study examining how HMB influences collagen I fibrillogenesis. There are several major concerns with this work: First, the study is entirely in vitro and non-biological: it is essentially chemistry. Importantly, the concentrations tested (0.025 M and 0.25 M) are far higher than plasma or tissue levels of HMB achievable through supplementation, making the findings biologically irrelevant. Second, with only two concentrations, the study does not provide an appropriate dose–response curve to determine whether the effect is gradual or threshold-dependent, which further lowers the credibitiy of the observation. Thus, given the minimal data, the discussion of mechanistic interpretations and conclusions is unsupported. Besides, the paper requires a much clearer explanation of statistical methods and the number of replicates.  

Author Response

We sincerely thank the Reviewer for the critical evaluation of our manuscript. We acknowledge the concerns regarding the in vitro nature of the study, the concentrations applied, the absence of a full dose–response, and the need for clearer methodological explanation. We have revised the text to clarify the aim of this work as a mechanistic, proof-of-concept in vitro study rather than a direct simulation of physiological conditions, justified our choice of concentrations, and expanded the description of the statistical analysis and replicates. Detailed responses to each point are provided below. For clarity, line numbers refer to the revised version (no tracked changes).

  1. First, the study is entirely in vitro and non-biological: it is essentially chemistry. Importantly, the concentrations tested (0.025 M and 0.25 M) are far higher than plasma or tissue levels of HMB achievable through supplementation, making the findings biologically irrelevant.

Thank you for highlighting this serious concern; we appreciate the opportunity to address it. We share the reviewer’s reservations and have updated the text to respond directly to this point. Our study is intentionally an interaction-focused, molecular model designed to probe how HMB influences type I collagen assembly under tightly controlled pH/ionic conditions. We now make this explicit in the Abstract and Introduction and have removed in vitro from the title to keep it concise, while clearly stating the in vitro nature within the text. We also clarify that the work provides design inputs for collagen-based biomaterials/formulations, where additive levels during processing can be formulation-level rather than physiological (Lines 121-126).

  1. Second, with only two concentrations, the study does not provide an appropriate dose–response curve to determine whether the effect is gradual or threshold-dependent, which further lowers the credibitiy of the observation. Thus, given the minimal data, the discussion of mechanistic interpretations and conclusions is unsupported.

 We thank the Reviewer for this important comment. We fully agree that testing only two concentrations does not allow us to construct a full dose–response curve or to distinguish gradual from threshold-dependent effects. Our aim in this preliminary in vitro study was to explore whether HMB, as a small bioactive molecule, can directly modulate collagen self-assembly under simplified experimental conditions, rather than to map a physiological dose–response. In line with this, we have revised the Introduction (Lines 121-126) and Conclusions (Lines 532-546) to present the work as a proof-of-concept study more clearly. The conclusions are now phrased in more balanced terms, highlighting the concentration-dependent effects observed while avoiding over-interpretation of the underlying mechanism. Additionally, in the Study Limitations, we explicitly note that intermediate concentrations were not tested and that future studies should address this gap to better define whether the response is gradual or threshold-like (Lines 519-526).

 3. Besides, the paper requires a much clearer explanation of statistical methods and the number of replicates.

 We thank the Reviewer for this important remark. We clarified in the Materials and Methods section that the experimental unit in all inferential analyses was the biological replicate (n = 3 independent preparations per group).

Statistical analysis (Lines 499-503): “For all inferential analyses, the experimental unit was the biological replicate (n = 3 independent sample preparations per group). Technical replicates (≥30 fibrils or spectra per group) were averaged within each biological replicate to avoid pseudoreplication. Only these biological means were entered into the ANOVA/Tukey tests.”

Sample Preparation and Atomic Force Microscopy (AFM) Analysis (Lines 455-456): “Fibril-level measurements were averaged within each biological replicate, and only these replicate means were used in the statistical analyses”

Fourier Transform Infrared (FTIR) Analysis (Lines 473-475): “Spectra were averaged within each biological replicate, and these biological means (n = 3) were used for statistical analyses.”

Apart from that, we have revised the Statistical Analysis subsection to make the analysis applied more clear (Lines 491-493): “A one-way analysis of variance (ANOVA) was used to evaluate changes in both collagen fibre morphology (width, height, D-banding) and FTIR-derived spectral characteristics (Δυ, AmideIII/1450cm-1 ratio and Amide I secondary structure).”

Reviewer 5 Report

Comments and Suggestions for Authors

The study by Izabela Świetlicka et al. is devoted to investigating fibril formation from type I collagen under the influence of β-hydroxy-β-methylbutyrate (HMB) in an in vitro model. HMB is a natural compound produced in a human body; it is used both as a dietary supplement and as a component of medical formulations for wound healing. This compound is also beneficial for individuals with muscle atrophy. HMB has a favorable safety and efficacy profile. Despite its widespread use, the precise molecular mechanisms underlying its modulatory effects have not been thoroughly studied. Understanding these mechanisms enables the development of individualized therapeutic regimens. Therefore, I consider the authors’ work highly relevant and of great interest to readers..

The manuscript consists of standard sections, is properly structured, and the content corresponds well to the title. It cites 60 papers, almost one third of which are from the last five years. To test their hypothesis, the authors conducted an in vitro study examining fibril formation from collagen under the influence of two concentrations of HMB. Fibril morphology was analyzed using atomic force microscopy, and molecular structure was investigated by FTIR spectroscopy.

The study design, the choice of methods, and the data processing appear adequate. The methods are described in sufficient detail to allow replication of the experiments. The manuscript includes three figures and three tables, which accurately represent the main results and are easy to interpret. The authors’ conclusions are supported by the presented evidences. The introduction convincingly substantiates the problem statement, and the discussion is written in an engaging and scientifically sound manner.

I highly appreciate this work and believe it contributes to the understanding of fibrillogenesis mechanisms, offering scope for further research and exploration. However, during the review, I formulated several comments and suggestions for the authors’ consideration.

  1. In the keywords, please spell out the abbreviation AFM as “Atomic Force Microscopy.” Also, I suggest including "fibrillogenesis" in the keywords while removing "Amide I," as it pertains more to data interpretation details rather than the main topic.
  2. In the Limitations section, the authors provide a convincing and detailed justification of the study’s limitations and prospects for further research. However, this section appears overly detailed. I recommend retaining only a few general statements here and transferring specific discussions related to Amide bands, particular concentrations, and similar details to the Results and Discussion sections. Overall, the Limitations section should be shortened by approximately two to three times..
  3. The same applies to the Conclusion section, which should be expressed in more general terms. For instance, I suggest moving the following detailed passage to the Discussion section: "... accompanied by a redistribution of Amide-I sub-bands toward α-helical/β-sheet-like contributions, a spectral shift of Amide II towards higher wavenumber, growth of the carboxylate band near ~1400 cm−1, and the emergence of a distinct band at 1114 cm−1 indicative of interfacially bound β-hydroxycarboxylate. Importantly, the Δυ (Amide I–Amide II) criterion remained below 100 cm−1 across conditions, supporting maintenance of the triple-helical backbone despite supramolecular reorganisation." The Conclusion should focus on summarizing the main findings in broad terms, emphasizing their significance and potential implications, rather than the detailed spectral analysis. This adjustment will improve clarity and coherence, aligning with best practices in scientific writing.
  4. Regarding Tables 1, 2, and 3, please indicate the number of samples used to calculate the mean and standard deviation, as well as the number of parallel measurements performed.
  5. In Table 1, the reported precision of size determination appears too high. Considering the use of an atomic force microscope with a probe diameter of approximately 10 nm, the lateral measurement accuracy is about 1 nm, and the vertical accuracy is around 0.1 nm. Therefore, values such as 358.28 ± 57.809 seem excessively precise. They should be rounded to 358.3 ± 57.8 or, at best, 358 ± 58. For height measurements, the error may be smaller; hence, values like 30.28 ± 4.120 should be presented as 30.3 ± 4.1. Please revise the entire table accordingly.
  6. In Table 2, please adjust the reported mean and standard deviation to consistent precision levels—for example, 83.2 ± 1.9.

Minor comments

  1. For all tables and figure captions, please provide explanations for all abbreviations used.
  2. Please include a reference to Figure 3 in the main text prior to the figure itself..

Author Response

We sincerely thank the Reviewer for the very positive and encouraging assessment of our work. We are grateful for the constructive suggestions regarding the refinement of keywords, the conciseness of the Limitations and Conclusions sections, the precision of reported data in the tables, and the clarity of figure references and captions. We have carefully addressed all these points in the revised manuscript, and detailed responses to each comment are provided below. For clarity, line numbers refer to the revised version (no tracked changes).

  1. In the keywords, please spell out the abbreviation AFM as “Atomic Force Microscopy.” Also, I suggest including "fibrillogenesis" in the keywords while removing "Amide I," as it pertains more to data interpretation details rather than the main topic.

We thank the Reviewer for this valuable suggestion. The keywords have been revised accordingly.

  1. In the Limitations section, the authors provide a convincing and detailed justification of the study’s limitations and prospects for further research. However, this section appears overly detailed. I recommend retaining only a few general statements here and transferring specific discussions related to Amide bands, particular concentrations, and similar details to the Results and Discussion sections. Overall, the Limitations section should be shortened by approximately two to three times.

We appreciate this guidance. The Study Limitations section has been shortened to highlight only the general constraints (lack of intermediate concentrations, absence of kinetic analyses).

  1. The same applies to the Conclusion section, which should be expressed in more general terms. For instance, I suggest moving the following detailed passage to the Discussion section: "... accompanied by a redistribution of Amide-I sub-bands toward α-helical/β-sheet-like contributions, a spectral shift of Amide II towards higher wavenumber, growth of the carboxylate band near ~1400 cm−1, and the emergence of a distinct band at 1114 cm−1 indicative of interfacially bound β-hydroxycarboxylate. Importantly, the Δυ (Amide I–Amide II) criterion remained below 100 cm−1 across conditions, supporting maintenance of the triple-helical backbone despite supramolecular reorganisation." The Conclusion should focus on summarizing the main findings in broad terms, emphasizing their significance and potential implications, rather than the detailed spectral analysis. This adjustment will improve clarity and coherence, aligning with best practices in scientific writing.

We agree with the Reviewer’s suggestion. The detailed spectral description was transformed and the Conclusion was rewritten to provide a concise summary of the main findings in general terms, emphasising the significance of HMB as a modulator of collagen fibrillogenesis (Lines 532-542).

  1. Regarding Tables 1, 2, and 3, please indicate the number of samples used to calculate the mean and standard deviation, as well as the number of parallel measurements performed.

We have updated the captions of Tables 1, 2, and 3 to specify that the results are based on three independent biological replicates per group, with a minimum of 30 fibrils measured per AFM sample and triplicate spectra per FTIR sample.

  1. In Table 1, the reported precision of size determination appears too high. Considering the use of an atomic force microscope with a probe diameter of approximately 10 nm, the lateral measurement accuracy is about 1 nm, and the vertical accuracy is around 0.1 nm. Therefore, values such as 358.28 ± 57.809 seem excessively precise. They should be rounded to 358.3 ± 57.8 or, at best, 358 ± 58. For height measurements, the error may be smaller; hence, values like 30.28 ± 4.120 should be presented as 30.3 ± 4.1. Please revise the entire table accordingly.

We thank the Reviewer for pointing this out. Table 1 values have been rounded to reflect the actual measurement accuracy (to 0.1 nm). Standard deviations were adjusted to the same decimal places for consistency.

  1. In Table 2, please adjust the reported mean and standard deviation to consistent precision levels—for example, 83.2 ± 1.9.

We revised Table 2 accordingly: all values are now reported with consistent decimal places (one decimal for Δν, three decimals for Amide III/1450 ratio).

  1. Minor comments
  • For all tables and figure captions, please provide explanations for all abbreviations used.

All table and figure captions have been revised to include explanations of abbreviations (e.g., HMB, AFM, FTIR, SD).

  • Please include a reference to Figure 3 in the main text prior to the figure itself.

A reference to Figure 3 was added in the main text before the figure presentation (Line 344).

Round 2

Reviewer 1 Report

Comments and Suggestions for Authors

the authors adequately answer the questions and improve the article and it can be accepted

Reviewer 4 Report

Comments and Suggestions for Authors

In its current form, the paper has resolved most of the technical issues. However, concerns remain regarding the scope and impact of the work.

Reviewer 5 Report

Comments and Suggestions for Authors The authors have done a good job and have taken my comments into account to the best of their ability. I am completely satisfied with their comments. The article may be accepted for publication.